# Community-Based Approaches in the Construction and Management of Water Infrastructures among the Chagga, Kilimanjaro, Tanzania

**Valence M. Silayo** [1] **and Innocent Pikirayi** [2,*]

1    School of Education and Human Development (SoEHD), Dar es Salaam College (TUDARCo), Tumaini University, Coca-Cola Road, Plot No. 10, Mikocheni Industrial Area, Dar es Salaam P.O. Box 77588, Tanzania
2    Department of Anthropology, Archaeology and Development Studies, Faculty of Humanities, University of Pretoria, Lynnwood and Roper Street, Pretoria 0028, South Africa
*    Correspondence: innocent.pikirayi@up.ac.za

**Abstract:** Water management among the Chagga people of Kilimanjaro has involved community collaboration in the construction, ownership and management of water infrastructures. Since the second half of the second millennium AD, the Chagga settlement on the lower slopes of Mt Kilimanjaro significantly transformed the landscape to reflect an agrarian society characterised by decentralised forms of socio-political and economic organisation. Such organisation involved conception, construction, and post-construction management of water distribution systems, synonymous with high levels of socio-political complexity. The study employs ethnography and archaeological surveys to document the construction of water infrastructures on the lower slopes of Mt Kilimanjaro. An ethnographic survey among Chagga elders generated primary data on water furrow construction. This information was then used to aid archaeological surveys in mapping irrigation furrows (*mfongo*) in the lower slopes of Mt Kilimanjaro. The ethnography also provided data on how Chagga chiefs and clan leaders governed the construction, use and maintenance of water infrastructures in the past. Such approaches highlighted Chagga lived experiences of traditional irrigation technologies and infrastructures and how these developed a complex agrarian society. Results show that community collaboration was key in the management of water infrastructure vital for their home gardens, and this sustained Chagga society for centuries.

**Keywords:** Chagga; Kilimanjaro; water infrastructures; irrigation; furrows (*mfongo*); furrow construction; water management; home gardens; chiefdoms; clans

## 1. Introduction

Mt Kilimanjaro is Africa's highest mountain located in northern Tanzania, in eastern Africa. Comprising five ecosystem zones, it has the cultivation zone on the lower slopes (800–1800 m above sea level), the montane forest (1800–2800 m), heather-moorland (2800–4000 m), the alpine desert (4000–5000 m) and the arctic summit, over 5000 m [1–3]. The cultivation zone is home to a number of ethnic groups, comprising the pastoral Maasai on the northern and north-western slopes, the Ongamo on the eastern slopes, and the Chagga agro-pastoralists on the southern and western slopes of the mountain [4]. The largest of these ethnic groups, the Chagga, have traditionally lived on these slopes as well as on the eastern part of Mount Meru for at least four centuries, exploiting these vast mountain ecosystems [5–7]. Their landscape can be divided into two—the highlands and the lowlands—the high and lowlands performed different but related functions [5]. For the Chagga, land provided opportunities for various development in all their spheres of life. To attain their daily subsistence needs, the Chagga began transforming the native mountain forest. Trees that provided fodder, fuel and fruit were retained. In contrast, the less useful ones were eliminated and replaced with new tree and crop species [8]. The

resultant cropping system and subsistence economy resulted in a high population in the region. These highly fertile ecosystems have generated considerable agricultural wealth for the Chagga, based on indigenous traditions of extensive irrigation systems, terracing of the mountain slopes, and organic enrichment of soils [9]. In this paper, we engage contemporary Chagga communities on ancient Chagga traditions using ethnographic and archaeological approaches, to understand water distribution systems on the lower slopes of Mt Kilimanjaro and how these are connected with the origins and development of complex social systems around the mountain [10]. The Chagga still use these water infrastructures [11–13] since these are key to their home garden (*kihamba*, or *vihamba*) system [8]. The latter is a socio-economic and ritual space that is central to Chagga identity and culture, and which defines their worldview, including life and death (water infrastructures, home gardens). Today, the Chagga neither construct new irrigations infrastructures nor follow the ancient post-construction rituals. It is the memory and reverence to the art and skills their ancestors invested in furrow construction that still live with present day communities and that attest to their origins and development as a complex agrarian society in eastern Africa. One may compare the Chagga to the advanced chiefdoms, given their complex socio-political organisation [14,15]. Listening to the contemporary Chagga narrating the art and skills of their ancestors, as our ethnographic data attests, one discerns how their achievements in the past are critical in shaping their present. Narratives on the nature and systems of water distribution and management give the Chagga a deep sense of ownership of these water infrastructures, which they maintain and preserve.

Sub-Saharan Africa is home to "islands of intensive agriculture", involving innovative forms of water management [16,17]. Such ecosystems have hardly been considered within the context of how such societies developed complex socio-political systems [10]. The Chagga remain some of the few societies in Africa living in an environment characterised by intensive crop cultivation, whose development resulted in a highly complex agrarian system. Numbering about 2 million, they are regarded as the wealthiest ethnic group in Tanzania. Since 1500 AD, they developed complex social and administrative systems that were not too monarchical, but only organised at chiefdom level, comprising about 400 clans that managed an ever-increasingly dense and growing population on the slopes of Kilimanjaro [7,10]. Kilimanjaro's montane ecosystem allowed for the development of water infrastructures in the form of furrows (*mfongo*, *mifongo*) and water collection dams or reservoirs (*nduwa*), which supported agriculture sustainable practices for centuries [1,2,11–13].

## 2. Literature Review

The literature discussed in this section highlights historical and contemporary knowledge on Chagga water infrastructures and their importance in shaping the worldviews of the inhabitants of the lower slopes of Mt Kilimanjaro. While some archaeological work on these infrastructures has been conducted, (see, e.g., [10,16]), there remains limited knowledge regarding furrow (*mfongo*) construction and how such hydraulic engineering linked the natural water sources flowing from the rainforest, with Chagga home gardens (*kihamba*). Since the Chagga neither construct new irrigation infrastructures nor follow the ancient post-construction rituals, our primary research question is how did they do so? Although available historical accounts report on such hydraulic features and structures among the Chagga, e.g., [8,18,19], research gaps still exist on construction technologies and how communities were mobilised in water management. Thus, we further ask the extent to which more systematic ethnographic and archaeological surveys help fill in the gaps in existing literature.

The Chagga are divided into several chiefdoms, the most prominent ones being Kibosho and Keni-Rombo (Figure 1). They have always been entrepreneurial and adaptive, a common cultural trait strongly associated with the lower slopes of Kilimanjaro [8,18]. Monumental structures, such as underground bolthole, and water management features such as furrows, attest to conflict and warfare between some Chagga chiefdoms, feats of hydraulic engineering and clan or communal organisational capacities within the broader mountain region [19,20].

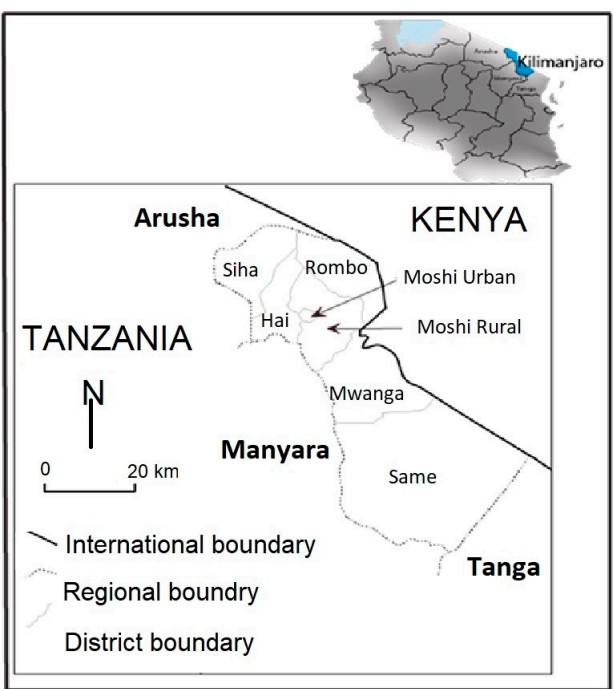

**Figure 1.** The Kilimanjaro region showing some of the places mentioned in this paper (image courtesy of Valence Silayo).

Human settlement history of the Kilimanjaro region remains scanty. Evidence from western Kilimanjaro suggest a human presence since Neolithic times [21], while pottery from Marangu, east of the mountain, points to Iron Age farmers during early first millennium AD. Elements associated with ancestral Chagga settlement such as irrigation and associated water infrastructures date some 400–500 years ago [22]. Recent work [10] provides evidence of complex forms of settlement incorporating furrow construction and water management dating from the mid-fifteenth century AD. This is supported by early visitors to Kilimanjaro [4,19,23,24]. Bishop Hannington recorded an account from Mangi Mandara of Moshi, who claimed ancestry going back fourteen generations, which, according to oral traditions would date Chagga settlement in the Kilimanjaro region to the mid-first millennium AD [24].

Land-use among the Chagga is characterised by intensive smallholder production of subsistence and cash crops. The "kihamba system" or home gardens enabled farmers to sustain the production of food using water harvested from the mountain and liquid manure and mulch [8]. The home garden is typically a complex multi-cropping system that has evolved over many centuries through a gradual transformation of the southern and western slopes of Mt Kilimanjaro. It is also an explicit example of agricultural resilience as it integrates numerous multi-purpose trees and shrubs with food crops and animals without a specific spatial arrangement. The system still comprises four layers of canopies: food crops such as maize, millet and beans, coffee, bananas, and trees, maximising the use of limited land and increasing resistance against droughts and pests while ensuring environmental protection. Chagga agriculture intensified during the nineteenth century following the introduction of coffee beans and other exotic crops [25,26]. Millet and banana are central to Chagga subsistence practices, culture and identity. Sir Theodore Morison remarked that the Chagga grew a "sort of canary seed used to brew local beer, and also grew a variety of bananas—for food, cattle, and some were dried up to make flour, and yet another one mainly used or eaten with meat" [27]. Long-distance caravan trade between the coast and the Great Lakes region in the interior during the early decades of the nineteenth century contributed to the rising power of chiefdoms in the eastern African hinterland [28–30]. A caravan was composed of various ethnic groups and with a substantial number of people. The Chagga were strategically located to refuel such large groups of people with essential supplies. Chagga chiefs had to mobilize resources such as food, water, ivory, and slaves to

meet the needs of the traders. The caravan trade intensified conflict between chiefdoms. Over time, some chiefdoms dominated the trade, and would raid neighbouring chiefdoms to procure slaves and other commodities [10].

Early Chagga settlements were usually located on rivers and streambanks, with fast-moving water flowing toward the plains. Such locations served a dual purpose; security and easy access to water. Riverbanks served as entry and exit points of boltholes and also supplied water for domestic use [10,20]. The Chagga would utilize these numerous rivers and streams flowing from the mountain to develop water infrastructures to support an intensive agricultural subsistence economy. The forest zone, between 1800 and 2800 metres above sea-level, provides most of the water used in the cultivation zone, which, according to Ramsay [31] is "a network of irrigation channels and furrows". The furrows (*mifongo*) run for several distances, sometimes up to 15 kilometres, tapping water from different streams and rivers flowing from the mountain [10]. Where residents living on the lower slopes of Mt Kilimanjaro obtained the knowledge of *mfongo* construction from remains uncertain. The idea might have been borrowed from neighbouring groups, such as the Engaruka north of the mountain or the Kamba in the east, in what is now modern Kenya [18,31]. The Engaruka and the Kamba may have practised irrigation much earlier than the Chagga, who only started irrigation during the first half of the seventeenth century AD [10]. This coincides with the formative phase of Chagga settlement in Kilimanjaro, between the fifteenth and seventeenth centuries, characterised by numerous wars between chiefdoms and accompanied by the building of war ramparts. Such battlements include underground tunnels, ditches and stone fortifications [18,26].

Water furrows in the Kilimanjaro region have been extensively described [10,18,32]. Historians such as Stahl [33] and early visitors such as Johnston [4,23] described the furrows and water flow from mountain streams to the fields as a spectacular feat of engineering. In the past, streams and rivers such as Kikafu, Weruweru, Kikuletwa, Mue, Karanga, Rau and Wona had running water throughout the year and water furrows were connected to these. This paper also highlights the complex interconnectedness between social organization and maintenance behind these structures. The physical landscape poses considerable obstacles that required sophisticated planning of water infrastructures from the gorges to the banana and settlement groves [10,12,13]. This also explains the ritual practices and seriousness Chagga clans attached to water management [10,11]. Since it is costly to construct a furrow and channel water through them, the Chagga adopted very strict rules to ensure sustainability, safety and cleanness of the water flowing therein.

## 3. Research Methods

A desktop literature on the history of the Chagga and human settlement on the lower slopes of Mt Kilimanjaro provides the background context of this study. We then employed ethnographic and archaeological surveys to locate and map cultural markers left by ancestors of the present Chagga. Markers such as furrows and water reservoirs, give insights into past and present water availability, distribution and use. Ethnographic enquiries were conducted to collect details on furrow construction. Although only a few informants are mentioned by name in this paper, our ethnographic surveys in the Mt Kilimanjaro region are informed by 90 informants, interviewed since 2013, all recruited by way of the snowball sampling, whereby participants were asked to assist us in identifying other potential subjects. The method was preferable because it worked well with the survey techniques employed in the study, such as archaeological surveys, which also required asking participants where sites were located. Elders of advanced age were identified and regarded as key informants. Those we interviewed were requested to lead us to extinct or existing irrigation furrows in their areas and beyond. The exercise included locating the source of furrows—normally mountain streams or rivers—and any other structures relating to past or present irrigation systems, such as reservoirs. This approach was also an exercise in cultural as well as archaeological mapping of the irrigation structures (water infrastructures) identified. The survey methods employed in this research also allowed for assessing the landscape more dynamically, (see,

e.g., [34]). It is evident that features and structures connected with past irrigation furrows as well as associated settlements disappear for various reasons, in response to natural and cultural formation processes. Through ethnographic as well as systematic archaeological surveys (see details below), one is able to identify some of these ancient or historical features, structures and cultural landscapes including furrows, reservoirs, and terraces [35]. Studies on some communities in eastern Africa such as the Pokot, the Marakwet, the Pagasi, the Baringo, the Sonjo, the Taita and the Pare have revealed evidence of intensive agriculture in the past, which employed either non-intensive or intensive irrigation [12,13]. Similarly, among the Chagga, is evidence of ancient water management systems to irrigate their fields, which they still utilise today, since it is central to their economy and social organisation. An analysis of these structures with the help of ethnographic data allows for an understanding of how ancient water infrastructures among the Chagga facilitated the development of complex forms of organisation [10,36–40]. In this paper, the term water infrastructures in a traditional sense, in reference to infrastructures related to agriculture, such as water furrows, water reservoirs, irrigation systems, etc. According to ethnographic and historical research, these infrastructures have consequences for social and political organization, and for the Chagga this is quite evident [10–15,33,36].

Archaeological surveys identified and mapped various irrigation structures and layout in three districts: Rombo, Moshi Rural and Hai (Figure 1). These surveys focused on the construction of water furrows in relation to the associated physical landscape. They had two aims; first, to identify and map the distribution of accessible irrigation furrows and the water courses they originate from; and, secondly, to understand the physical landscape in relation to the rivers as well as the intended distance from such water sources to the villages. Such surveys also offered an opportunity to document the technology involved in the construction of particular water infrastructures. The research area, however, has no open places like glades that are fairly easy to survey by way of systematic walking along available terrain. Much of the landscape where the Chagga homes are comprise built-up areas and associated home gardens [5,7,8]. Given the extensive area selected for research and the hilly topography of the area adjacent to the home gardens, it was only possible to survey selected parts of that universe. Therefore, to cover as much area as possible, a pedestrian survey method was preferred, which allowed the research team and the informants to traverse the terrain following particular water furrows. Most of the surveys commenced at the source/water intake points, aligned to furrow structures as much as possible and proceeded downhill towards the local villages.

In evaluating the ethnographic data, we employed technopolitical approaches to understand how Chagga chiefs and clan leaders governed water infrastructures, ensuring their continuity to the present in relaying a vital resource to their home gardens. Chagga pasts are best understood using sociotechnical approaches that emphasize ways in which their water infrastructures mediate or mediated the relationships between people and social institutions, and between people and the montane environment. Such approaches are phenomenological, drawing on a community's lived experiences of their worldview. They are also vita in understanding local, indigenous perspectives around management of traditional irrigation technologies and infrastructures and how complex social and political organization developed around them [12,13,16]. With substantial evidence of anthropogenic modification of the montane ecosystem over centuries, the lower slopes of Kilimanjaro bear testimony to the evolution of societies through use and control of water resources [29,36–39]. We contend that Chagga water infrastructures were an investment to cope with continuous environmental change and future uncertainty over water supply and use on the lower mountain slopes of Kilimanjaro. This comes from our interpretation of the ontologies and politics of water, as informed by ethnographic data. Water infrastructures premised continuous survival of the Chagga. Thus, this paper goes beyond a discussion of water infrastructures as evidence of complex Chagga pasts, demonstrating the intersection between the natural and cultural environment, and contemporary relevance. Such connections between humans and places, settlement and ritual have always been informed by cultural landscape approaches [41].

## 4. Results

From the ethnographic data collected, our informants highlighted different roles played by respective individuals in the community hierarchy regarding water ownership, furrow construction and management. Although Chagga chiefs (*mangi*) had the final say regarding various issues on water such as permitting the cutting of furrows from a specific river to officiating and mediating on water disputes that water boards or committees could not resolve, they did not own water distribution structures. According to Mr Tobias Milioni Mushi from Kibosho: "the idea begins with an individual, mostly a clan furrow engineer", this implied a necessity for that furrow and consultations with villagers regarding the specific needs and uses of the water from the new furrow were thus required. This is also evidenced by the fact that furrows are not owned by specific families, but rather, particular clans or collectively by the entire village. All the furrows identified during the study are under the ownership of specific clans and or villages, e.g., the Mamba furrow in Machame, which is owned by Mamba village. This arrangement ensured rights to water access to every individual in that and other villages. According to Mr Tobias Milioni Mushi, the current chairperson for Orosise furrow (pers. comm), after internal consultations among the immediate users of the intended furrow, a formal request is submitted to the chief (*mangi*) by the clan through the chief's local representative (*mchili*), to allow the clan/village members to cut a new furrow.

Ethnographic surveys also revealed that after construction, furrows were placed under a management committee led by elders. The committee was a custodian for an asset owned collectively by the clan. In consultation with the *mangi*, the committee would take responsibility for the daily operation of the furrows, directing and redirecting water depending on need and availability. A strict schedule was maintained when releasing water to irrigate fields, feeding the animals and for household consumption. Every clan member was obliged to report a damaged or leaking furrow, thus protecting such infrastructures was a collective as much as it was an individual responsibility In the event of damage or leakage, a furrow elder would announce during the evening a message known as "*Ole lo mfongo*" (call for furrows). Regardless of personal routine, everyone was expected to attend to this call the next morning.

Each Chagga chiefdom had a specific clan that owned the art of furrow engineering with the Mbokomu chiefdom at the core of this technology. One of their leaders, Mangi Mlatie, was a revered furrow surveyor. Collectively, these clans are known as 'Wakomfongo'. Surprisingly, not every member of Wakomfongo was an expert in furrow engineering. The vision to start and lead furrow surveying was conceived individually and the person who had the vision (usually manifested through a dream) would direct the work. In some instances, the idea came from other clan members. The clan would then convene and elect someone to direct the project. To get authorisation, the furrow surveyor would present a gift from the potential users of the furrow to the Mangi, either in the form of a *ndafu* (fattened he-goat) or *mbege* (traditional beer). The Mangi would assess the request including the available human resources and if satisfied, consents to the project. This tradition existed since time immemorial. The blessings from the Mangi were not enough to warrant the beginning of the furrow construction. A series of rituals, ceremonies and observations by the engineer and furrow users would follow. Mostly, such supplications and sacrifices were directed to the latest deceased elder of the clan, who then forwarded the request to the ancestors using set protocols. Oral accounts of these ritual traditions and observations differ, although some of them overlap. A conversation with Mr Anasa Mrema reveals that two immediate actors after the assent of the Mangi—the furrow chief engineer and clan elder—must observe a prayer vigil until they receive a signal. The signal, which comes in the form of a stretch of red ants from the home of the furrow engineer or elder of a furrow-owning clan or the home of the head of the clan, was interpreted as assent of the project from the ancestors. Members of the clan were expected to observe purity by abstaining from sexual intercourse and other profane behaviour, including blasphemy against the ancestors, unnecessary conflict and family squabbles. These observances would stay in place until

the signal is given, failure of which the project is postponed. Some clans would continue to observe this ritual until the construction of the furrow is completed and commissioned. In the event the construction project is completed but water fails to flow in the furrow, clan members took this very seriously, with the possibility that perpetrators who failed to observe the rituals were persecuted or punished (Aleangusiyo Oto Mushi, pers comm). These rituals speak to Chagga worldview, where water is regarded as pure, spiritual and harmless. It was a resource which mobilised the community, who never abrogated responsibilities related to furrow construction, maintenance or repair. This was sustainable in terms of the use "home-grown" resources, including clan participation. This is how social, cultural and political relationships developed around water management in the lower slopes of Kilimanjaro.Alongside ethnographic surveys, archaeological surveys were targeted towards specific areas of the mountain with both known and not so well-known water infrastructures. In Kibosho, surveys along the Isie River identified four furrows: Manga'tu, Mmasi, Orosise and Miroshi. In Machame, two furrows were identified along the Kikafu River: Uwia, and Mamba, while along Namwi River, Ngira furrow was documented (Figure 2).

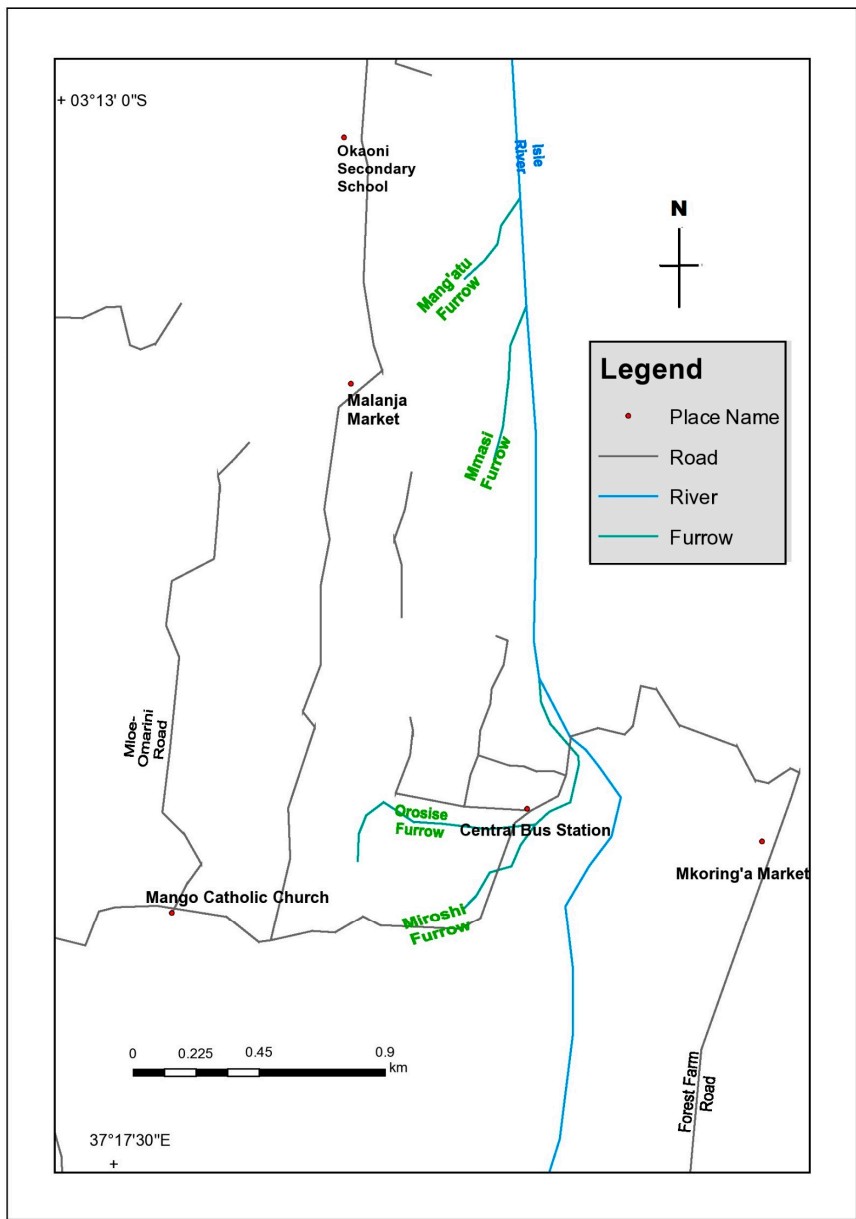

**Figure 2.** Irrigation furrows along Isie River, Kibosho (image courtesy of the GIS Laboratory, Stella Maris Mutwara University College).

In Marangu and Kilema we observed and recorded several furrows running alongside different rivers. Surveys in Machame revealed the difficulty inherent in mapping and constructing water channels for irrigation on the part of Chagga ancestors. Chagga civil engineers utilised intense knowledge of the landscape, which they navigated from the intake point on the river to the required water supply points. Water flow velocity is another important consideration in the construction of furrows. However, the proximity of villages (the user end-points) to the source or intake point (the river) was not as essential as the terrain contour and the maintenance of the infrastructure. For example, Mamba is one of the longest furrows in Machame (Figures 2 and 3), stemming from the Kikafu River and with its initial intake lying in the Kilimanjaro Forest Reserve.

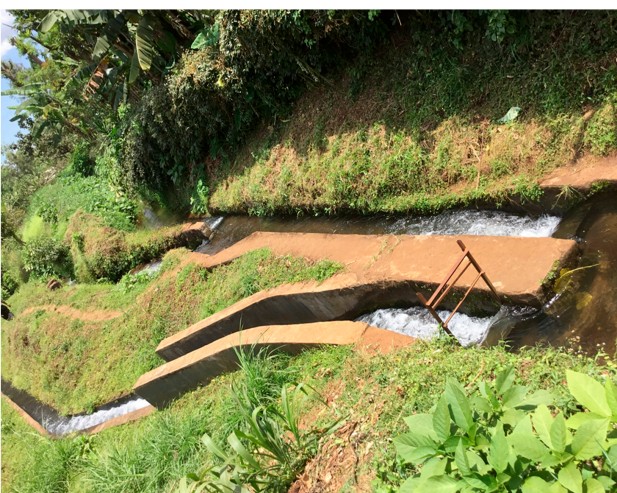

**Figure 3.** Mamba furrow complex in Machame, Kilimanjaro (image by Valence Silayo).

It runs for several kilometres from there to the villages. During an interview with Mr Aleangusiyo Oto Mushi (pers. comm), he indicated the furrow was initially intended to supply several villages. As such, a stable intake was essential but the deep Kikafu valley and the Machame hill contour could not provide a suitable point. We also observed that although there may have been physical obstacles imposed by the hill contour, engineers easily navigated around these. This is evidenced by furrows cut deep along the sides of gorges. Within such terrain, engineers were also able to reduce or increase water velocity in the furrows by manipulating the gradient (Figure 4). They also included reservoirs or water collection dams (*nduwa*) built a few meters up the intake point, which regulated the amount of water allowed into the particular furrows.

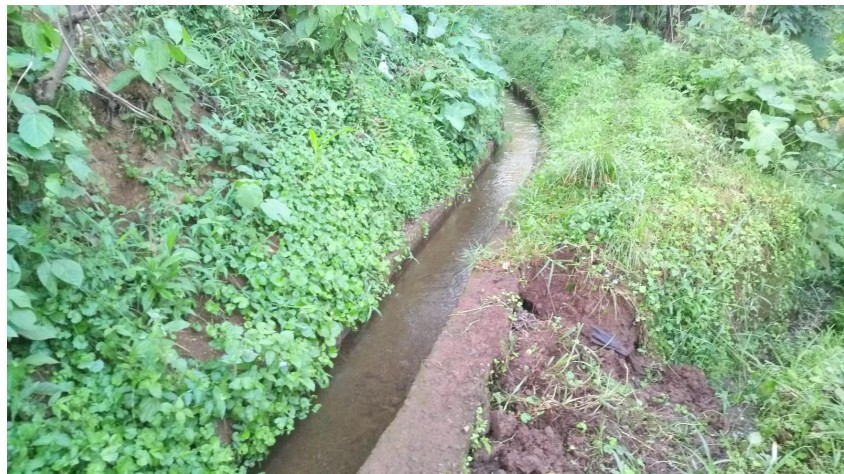

**Figure 4.** Furrows alongside the deep gorge of Ona River valley, Marangu (photo by Innocent Pikirayi).

Archaeological surveys reveal for the first time how Chagga engineers diverted water from mountain rivers, into furrows. This involved the construction of low embankment of stones and boulders at convenient points, where the water flow rate was minimal. Where valleys are deep, furrows run along hillsides for many kilometres until they reach the villages. Archaeological surveys also show some furrows originating from the deep forest higher up the mountain, meandering downslope towards the villages. Evidently, Chagga engineers meticulously mapped the landscape to allow for construction of furrows along gradients that would deliver water supplies to the villages located downslope with more than adequate velocity and quantity. At times, such villages had to be located some distance from the furrow points intake to allow the construction of appropriate water infrastructures to capture and divert the water from the rivers. Several furrows could be cut from the same river and one furrow could have numerous twigs or branches. In some cases, branches from the main channels extend to reach all parts of the village where each homestead is allowed to deviate a stream into the adjacent banana grove (Figure 3). Thus, the construction of water infrastructures in the lower slopes of Mt Kilimanjaro resulted in the emerge over time of complex irrigation systems among the Chagga, linking their home gardens (*vihamba*) with regulated and managed water flows from the rivers flowing from the rainforest.

At some point, furrows join to form one bigger furrow (Figure 2). From this point, secondary and tertiary channels (Figure 5) may spread, building a network of flowing water arteries [18].

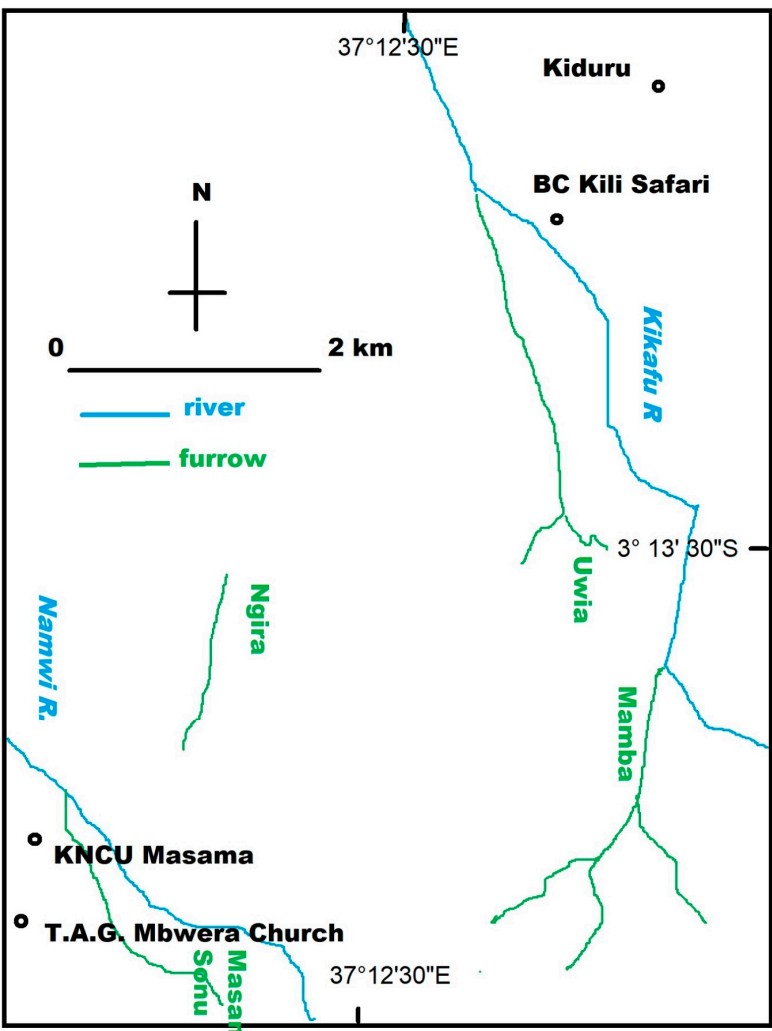

**Figure 5.** The distribution of furrows from Kikafu and other rivers, Machame (Prepared by the GIS Laboratory at Stella Maris Mutwara University College and modified by Innocent Pikirayi).

## 5. Discussion

The presence of complex irrigation systems among the Chagga implies the existence of an advanced socio-political organization in the interior of eastern Africa, among societies considered as small-scale in size [10]. Historical sources and early archaeological research among the Chagga give detailed accounts on how irrigation was used in all parts of Mt Kilimanjaro, especially the southern and western slopes. Masao [36] argues that *mfongo* technology and *nduwa* (water collection dams) are "as old as the Chagga people". On the same note, using oral traditions, Mattias Tagseth [38,39] points to Chagga irrigation as having been practised for a very long time. Chagga settlement on the lower slopes of Kilimanjaro required intricate integration with the rugged mountain terrain, whose biggest advantages were the rich volcanic soils and a reliable water supply from the rainforest. With these, the Chagga developed home gardens (*vihamba*), irrigated with water captured from the rivers and streams by way of *mfongo* and *nduwa* [18]. Over time, this created an intricate relationship between Chagga villagers and their physical environment. Water, land and livestock are the most important and highly valued Chagga possessions, a fact reflected in their numerous legends, songs and proverbs. The latter are used to guide the young and convey wisdom for future generations [42–44]. Water sourcing, usage and maintenance are central to Chagga legends and wisdom. These resources and practices have always been highly localized and managed, not through the authority of the chiefs, but rather, through village or clan leadership. Although the Chagga are no longer constructing water infrastructures, clan or village-level organisation is still relevant in the sustenance of these inherited structures and critically vital for their survival as a society.

The floruit of organization and logistics for furrow construction to the water management and maintenance of furrows demonstrates irrefutable elements of complex organisational capacities among the Chagga communities, organised along village and clan lines. Furrow construction was a community activity organised along clan lines. The highest level of political authority among the Chagga was the chief, who presided over several villages or clans in a particular area or region. This is where one locates the intricate political, social and ritual connections between Chagga chiefdoms, politics and water. From the ethnography, clans show a well-organized social system that facilitated and ensured a well-defined division of labour. Chagga clans played different but complementary roles in society; some were smiths, some rainmakers, some were *mfongo* engineers, while others were cattle herders or even beekeepers. Such clan division of labour made possible feats of furrow excavation manageable [45].

Furrow construction was performed by men and required enough and reliable personnel [9,18,40,46]. Their maintenance sustained Chagga agriculture [28,38,39]. Their deterioration would halt and affect the community's socio-cultural and economic development. Thus, despite conflicts among some chiefdoms over water, collaborative conservation of water infrastructures united various clans. According to Moore [47]:

> "To be able to utilise the perennial water resources, one had to tap water from the rivers at points higher up than the altitude at which the water should be used. To secure the supplies, one needed an agreement with the people living near the water source. Secure water rights imply political understandings among settlements through whose territory, or near whose land, the water passes. Water thus constituted a major reason for alliances between chiefdoms above and below and was a motive for amicable relations with chiefdoms on either side".

The furrow system involved very complex social systems, built over a long time [38,39,48]. It is striking, considering the number of people required to undertake such a breathtaking job. Many people required to construct and maintain the furrows indicate the level of social complexity and organization the Chagga had reached between 1400 and 1700 AD. Oral history in Kilimanjaro [18,45,49] show the Chagga clans persistently fighting among themselves and with other groups, for regional influence and dominance. Most remembered are the wars waged by Horombo of Keni, Mandara of Old Moshi and Sina of Kibosho [10]. Wars fought from the seventeenth to the first half of the eighteenth century

allowed very little time for the construction of water infrastructures; thus, the only time furrows could have been constructed without interruption was before this period.

Early visitors to Kilimanjaro describe these complex water distribution structures but did not witness the Chagga constructing them. Sir Charles Dundas [45] associates the development of furrow networks in Kilimanjaro with the cultivation of eleusine and intensification of other agricultural activities. This makes a convincing argument since most eleusine was cultivated during times of less or no rain, hence the need for alternative methods to irrigate the crop [50]. Masao [36], however, disagrees, arguing that eleusine was a major Chagga staple, a crop they would not risk adopting, the environment allowing it. Therefore, eleusine cultivation was apparently synonymous with the development of furrows. Irrigation systems in Kilimanjaro could be associated with the development of cereal agriculture, such as sorghum, between 1550 and 1800 AD [6]. Oral accounts from the study area echo Masao's [36] suggestion, pointing out that the local brew, *mbege*, was integral to the ritual and ceremonial ingredients required in the process of furrow construction and maintenance. *Mbege* is a social drink that accompanies all major Chagga gatherings, including meetings where decisions to construct new furrows were taken. We suggest the development of irrigation intensified eleusine production in Kilimanjaro.

According to Sunday [51], the coping, adaptive and transformative responses have long been characteristics associated with the vulnerabilities of the Chagga *mfongo* water management system. The nineteenth century caravan trade not only stimulated the Chagga farming system, but also their socio-political system [50]. The lower slopes of Kilimanjaro witnessed population growth and expansion of the agroforest area. Demand for water increased proportionally. Håkansson et al. [52], commenting on agricultural intensification in the nineteenth century, noted that communities living on long-distance trade routes needed substantial agricultural produce to feed the caravans and suggested that this demand might have prompted agricultural intensification along the caravan route. Earlier, Håkansson [53] argued that population densities in the Kilimanjaro region during the nineteenth century were low, and the region received sufficient rainfall. Regarding irrigation, he argues that the existing system expanded due to the nineteenth century caravan trade, which increased the demand for foodstuff. The Chagga also used water for millet (*Eleusine coracana*) cultivation. Apart from the Rombo region located on the leeward side of Kilimanjaro and thus a rain shadow area, the southern, windward slopes of the mountain witnessed eleusine cultivation during the dry period from June to October, using irrigation [51]. Furrow irrigation thus developed prior to the caravan trade, for the cultivation of millet and other crops as well.

Chagga markets radically transformed during the late eighteenth to the early nineteenth centuries following the introduction of regional markets and the increasing demand for agricultural produce for the caravans. The Chagga, as the main suppliers and providers of the caravan trade, experienced a shortage of agricultural produce and resorted to intensive agriculture to produce surplus food for the traders. By 1840, caravans to Kilimanjaro comprised large, diverse, multi-ethnic groups [54–56]. Although the Chagga had traded among themselves and with their neighbours for centuries, caravan trade intensified their irrigation systems, further enhancing a complex agrarian system. Chagga markets were run by women and protected by the Mangi and his men. In 1861, Karl Klaus von der Decken visited Old Moshi and described a market with about 500 women [54]. According to Tobias Milioni Mushi of Kibosho (pers. comm), every Mangi had a responsibility to protect these markets. While wars might have interrupted market days, these resumed immediately after the war [10]. The most revered warlord and state builder, Mangi Sina (1877–1897) of Kibosho, engaged in numerous wars in the Kilimanjaro region, while his people enjoyed long periods of peace and prosperity, farming and trading intensively [31,45].

The development of Chagga society was centred around water and irrigation. The hydrological engineering involved an intricate set of ritual and ceremonial events, linking Chagga chiefs, clans and villagers to the ancestral world. This attests to continuity and reverence of tradition, which attached considerable importance to water and associated infrastructures as vital components of the Chagga cultural landscape. Life among the Chagga

was and continues to be shaped by such infrastructures [38,39,57]. The investment, high-level organization, construction, use of these infrastructures was impressive and speaks to cultural and technological sophistication among the Chagga. Chagga surveyors used only a stick to plot the courses of furrows. They neither possessed nor required instrumentation for grading: their knowledge and skills were adequate for the water infrastructure they constructed [36]. Furrow alignment was done by way of visual inspection, the furrow excavated under rock or banked up [38,39]. It is such high level of local knowledge and manipulation of resources that defined early forms of Chagga socio-political complexity.

The Kilimanjaro region does not quite align with historian Karl Wittfogel's hydraulic theory of civilization that links control of water resources with the centralization of political power. In discussing the "absolute state", Wittfogel relates the importance of water to the rise of states, arguing that irrigation required substantial government representation and centralized state and economic control. He further argued that irrigation infrastructures such as canals are related to advanced societies [58,59]. While the Chagga water infrastructures are not comparable to some ancient Near Eastern and Asian complex societies that developed alongside major rivers, it is important to understand how chiefdom and state-level societies in sub-Saharan Africa interacted with water. The dynamic nature of the Chagga cultural landscape encouraged decentralized, rather than centralised political control [9]. Water in Kilimanjaro was pivotal for the growth of what Bender termed "sophisticated agrarian society" [18,31]. Thus, water infrastructures in Kilimanjaro exemplified a society with complex forms of community social organization allowing mobilisation and execution of large and complex structures such as irrigation furrows. According to Scarborough [60], the construction of water management infrastructures requires careful planning. This makes water infrastructures such as irrigation canals or furrows, a reliable measure of political power and state authority (see also [60,61]). Sir Charles Dundas [45] writing earlier than Karl Wittfogel about the development of furrow networks in Kilimanjaro said; "no small degree of regulation is necessitated and within the course of the furrow, the order must prevail". He attributed the whole process and management of the furrows to the institution of chieftainship and development of a steady organization. Although the furrows were developed and engineered by specific clans, construction work was overseen by chiefs. In this way, chiefdoms ensured smooth flow of authority and organisational leadership, vital in resolving disputes emanating from the use and maintenance of furrows. As a result, disputes arising from the use and maintenance of furrows and water rights during the past must have been few and well-handled [39,40]. The meticulous management procedures involved resulted in the drafting of community-by laws on water [9].

## 6. Conclusions

Both ethnography and archaeology enrich our understanding of Chagga clan histories and the complex agrarian developments centred around the home garden system, the *kihamba* that were served with water harvested from the upper mountain slopes. Since the second half of the second millennium AD, the lower slopes of Mt Kilimanjaro witnessed intensification of agriculture enhanced by the construction on the mountain slopes of water infrastructures in the form of water furrows and water collection dams, to support the *kihamba*. The link between home gardens and the water infrastructures attests to cultural continuity from the past, when the Chagga realised the need for a regulated supply of water for victualling the *kihamba*. The ethnographic evidence affirms that the construction of *mfongo* entirely relied on and promoted self-reliance, using local clan engineering expertise and raw material. This was possible through investment in community-based workforce and a management system that was highly adaptive to the needs of the local terrain or landscape and dictates of ritual and ceremony. The success of Chagga agriculture is attested by its sustenance of local and regional markets and maintenance of long-distance caravan trade during the eighteenth and nineteenth centuries. Chagga, home gardens were also a measure of agricultural flexibility, in an environment confronted with warfare or the demands of external traders.



The development of Chagga complex agrarian systems was made possible through the construction of water furrows, which conveniently conveyed water towards their home gardens. In this way, furrows provided the much-needed vital link between the rainforest or the fast-flowing rivers emanating from there, to Chagga homesteads. Such feats of hydraulic engineering were a successful way of pacifying torrential water flows, to usable, manageable flows for domestic use. This demonstrates successful Chagga efforts in transforming the lower slopes of Kilimanjaro, through the creation of such vital water infrastructures. Within the core social sciences, such developments are often associated with ancient complex societies, especially large kingdoms and empires. Ordinarily, the Chagga would be categorized as a "small-scale" society in eastern Africa, but this is a misnomer. Given that the Chagga were never a unitary state, the scale and extent of their water infrastructure would, however, categorise them as advanced chiefdoms around which they evolved complex socio-political and economic systems. We invite researchers working on the origins of "islands of intensive agriculture" to reconsider concepts that measure group size and territorial scale and reconsider how such "islands" may actually have developed as ancient state systems or kingdoms.

**Author Contributions:** Conceptualization, V.M.S.; methodology, V.M.S.; formal analysis, V.M.S.; investigation, V.M.S. and I.P.; resources, V.M.S. and I.P.; data curation, V.M.S.; writing—original draft preparation, V.M.S.; writing—review and editing, I.P.; project administration, V.M.S.; funding acquisition, V.M.S. and I.P. All authors have read and agreed to the published version of the manuscript.

**Funding:** This research was funded by African Humanities Program postdoctoral fellowship grant awarded to Valence Silayo (2018–2019) and the South African National Research Foundation (NRF) Competitive Program grant for Rated Researchers (CPRR) (Grant No. 105866) (2016–2019), awarded to Innocent Pikirayi. The APC was funded by the University of Pretoria.

**Data Availability Statement:** While some of the ethnographic data is unavailable due to privacy or ethical restrictions, the data used in this paper is archived with Figshare (https://figshare.com/, accessed on 22 December 2022) and is accessible upon request. The original GIS images of the mapping done for this research is archived at the GIS Laboratory at, Stella Maris Mutwara University College, Tanzania.

**Acknowledgments:** This study was undertaken as part of the African Humanities Program (AHP) and the project 'Approaching the Complexity of Great Zimbabwe' for which Valence Silayo and Innocent Pikirayi were the principal investigators respectively. Research permits were awarded by the Antiquities Department of the Ministry of Natural Resources and Tourism of Tanzania through Tumaini University, Dar es Salaam College Postgraduate Research Office. Particular gratitude is due to all the informants in Kilimanjaro and the local government officials. We are immensely grateful to the following Chagga elders for sharing their wisdom and profound knowledge with us during our fieldwork in October 2018: Tobias Milioni Mushi from Kibosho and the current leader of Orosise Furrow, who participated in numerous interviews on the history of Kilimanjaro; Anasa Mrema of Mbokomu, an expert on Chagga traditions and rituals; Yustina John Laswai, from Mkuu; Aleangusiyo Oto Mushi, from Machame; Eliasi Sebastian Ngoiya Laswayi; Rafael Mchau from Kidia, Old Moshi, and, Aurelia Clemence Lymo, from Marangu. The authors also appreciate the assistance from Pastor Enock Makundi of the Evangelical Lutheran Church in Tanzania who also heads the Northern Diocese Archive. We thank the University of Ghana and the Nordic Africa Institute for providing practical and logistical support as well as space for write up and discussion.

**Conflicts of Interest:** The authors declare no conflict of interest.

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
