# Peer review of "Community-Based Approaches in the Construction and Management of Water Infrastructures among the Chagga, Kilimanjaro, Tanzania"

_land, doi:10.3390/land12030570_

Round 1

Reviewer 1 Report

This work is an interesting anthropological study. However, it is not clear to us how the key informants were identified, how many there are, how representative they are, how they were chosen, how many were interviewed, what the survey design was, etc. Likewise, the methodology for georeferencing hydraulic infrastructures is not verified, so the figures leave much to be desired. It is understood that the main part is social development and its implications for production systems and their interaction with sociopolitical complexity. It can be seen that there is enough field and historical work to integrate a more complete article. My suggestion is that important methodological aspects be included in the article, such as those indicated, so that it ceases to appear as a merely anecdotal set of experiences. It is clear that the work has been done.

Reviewer 2 Report

Having read this manuscript, I find several part of the narratives difficult to comprehend, starting with the title. I have presented the feedback below to help the authors improve this manuscript.

Title: the title is too long, and the "the development of socio-political complexity among" part leave readers confused. I suggest a shorter title that is direct and concise – e.g.  Community-based approaches in the construction, ownership, and management of water infrastructures among the Chagga, Kilimanjaro, Tanzania."

Abstract: More methodological content is needed in the abstract. The abstract is not detailed, and this journal gives room for methodological details. The manuscript mentioned that "the study employs ethnography, archaeological surveys and GIS to document water infrastructures on the lower slopes of Kilimanjaro" (and that is all?). I suggest the authors at least describe the "how-to aspect of their methods and the specific results reached in the abstract. Three to five more sentences can do that. Otherwise, there is no logical connection to how they reached their conclusion.

Structure: The second section heading, "Environmental and Historical Background", makes a reader wonder, "for what?" the authors should be specific with their message in that caption as it leaves a reader unable to link section 1 with section 2. Be specific – "Environmental and Historical Background of….." (Kilimanjaro?). The caption makes for a poor section transition.

Result: Is the conclusion that "We conclude that community collaboration was key in management of the water infrastructure and by extension, agriculture, which sustained Chagga and chiefdoms for centuries" the main result of this study? What are the key new findings on this issue of water source and management?

Literature & reference: This is too poor. The introduction jumps right up to the Chagga landscape in Kilimanjaro with little to orient readers on the research's general state and then logically drives towards Kilimanjaro issues. The second section also dwells on the case. What is the foundational literature basis for this study? It is difficult to know because no effort was put into reviewing the literature. This is like simply taking readers into a case study with no literature orientation.

Moreover, it is not surprising that the manuscript lacks adequate references. The paper contains no 2022 citations and only one 2021 citation. Also, no 2021 citations and only three 2019 citations. To me (and I may be wrong), this suggests that the authors have not done enough to update the work in readiness for a journal of this type. Even in this journal, a lot has been published on the various aspects the paper strives to investigate.

Methodology: It is acutely problematic here. The research speaks about archaeological studies but does not detail how it was done or from where the secondary data was derived. It fails to orientate readers about what has been done on this work in the past. Some of the narratives are clear, but they are disjointed when viewed from the perspective of a study that should be reproducible.

Discussion and results: The results provide a narrative that is difficult to understand generally or specifically in the context of "Community-based approaches in the construction, ownership, and management of water infrastructures", which the title promises. The aspect of "…and the development of socio-political complexity" is quite unclear.

In general: this paper is too short of providing details of the claims within it. As an "Article" manuscript, this should be filled with details. How does one provide details of "Community-based approaches in the construction, ownership, and management of water infrastructures…" in a dozen pages? Furthermore, this includes three half-page-sized maps and two photos. In its current version, the manuscript lacks content. This is unacceptable because this is a qualitative narrative "ethnographic" manuscript.

Round 2

Reviewer 1 Report

I consider that the article is better argued and structured. I think that it meets what is necessary to be published

Author Response

We are very encouraged by the feedback from Reviewer 1 following our improvements of the paper. We paid attention to the English edits and attempted as much as possible to improve the conclusions so that they tie very closely with the results.

Reviewer 2 Report

Having read this manuscript again, I find it  still needing some more work in terms of details and addressing fundamental issues in the following ways:

Section 2 should simply explore literature on the subject. It, perhaps, should simply be titled "Literature review" and should actually explore the literature on the subject. Why the "research question" caption? The author is discussing case study and calls it literature review. No serious effort has been made to do actual literature that demonstrates a depth of knowledge on the subject.

Section 3: should simple be titled "Methodology" or "Research methods" and focus on explaining them. The "theory" added to the title is misleading.

These two key issues need to be addressed for the study to be readable in a scientific form.

Author Response

Reviewer 2 would like to see more substantive improvements on our paper, and we respect his/her comments, as s/he has indicated where these are needed. The reviewer is not as satisfied as Reviewer 1 on the improvements made thus far. In response, we addressed the concerns raised as follows:

For Section 2, we focused on the literature review as recommended and removed aspects that deal with the research questions. We also re-oriented our discussion to demonstrate, as the reviewer says, “the actual literature that demonstrates a depth of knowledge on the subject".

Regarding Section 3, we re-phrased it "Research Methods" and focused on explaining these further. We removed the term "theory". The research methods used in this paper are primarily two: ethnographic, which involves the use of oral interviews gathered from Chagga elders, and archaeological, which uses surveys of the terrain, to map water infrastructures. The desktop research used was a scoping exercise to map the literature already available that would provide further insights into the archaeology and ethnography. We have attempted to explain the methods used in the study in a direct, and less cryptic manner for readers who are non-archaeologists. Our use of relevant archaeological literature and related cultural landscape studies has attempted to communicate these towards a broader audience, beyond our own expertise. 

We are of the view that our research methods speak to an appropriate research design. Perhaps Reviewer 2 should have further suggested how this could be done to improve further the paper at this stage, as we attempted to address the very same concern during the first round of review.

We also contend that all the cited references relevant to the research. Our resubmission carried references that are well-aligned to the flow of the paper, and in doing so, we removed some references that we thought were not so relevant.

The paper as currently revised further clearly presents the research results and the conclusions reached are supported by the results.

We have addressed the English language and style, by cleaning up the grammatical and other errors as much as possible.

We however do not clearly understand what the reviewer says, and we quote; "these two key issues need to be addressed for the study to be readable in a scientific form". We are approaching our contribution from the Humanities background where we conceive the sciences differently. As such, we structure our research papers of this nature differently.